# Grey Correlation Analysis of the Durability of Steel Fiber-Reinforced Concrete under Environmental Action

**DOI:** 10.3390/ma15144748

**Published:** 2022-07-06

**Authors:** Yongcheng Ji, Wenwen Xu, Yichen Sun, Yulong Ma, Qiulin He, Zhiqiang Xing

**Affiliations:** School of Civil Engineering, Northeast Forestry University, Harbin 150040, China; yongchengji@126.com (Y.J.); xww202206@126.com (W.X.); syc202206@126.com (Y.S.); myl2001928@163.com (Y.M.); hql2000@126.com (Q.H.)

**Keywords:** steel fiber-reinforced concrete, environmental effects, freeze-thaw cycle, micro-structure, mechanics performance, grey correlation degree

## Abstract

The interface performance of steel fiber-reinforced concrete (SFRC) is a critical factor in determining mechanical properties and durability. The degradation of the concrete matrix and micro-structure interface is caused by environmental erosion, which shortens the service life of the structure design. Considering different volume contents of steel fiber (0%, 1%, 2%), the failure mechanism of SFRC under different environmental erosion conditions was studied through a laboratory test scheme. A total of six environmental factors are selected, including water, sodium chloride solution, sodium sulfate solution, dilute sulfuric acid solution, sodium hydroxide solution, and a freeze-thaw cycle. When subjected to different erosion concentrations and periods, micro-structure and axial bearing capacity deterioration laws are compared and analyzed. A durability equation related to fiber mixture ratio and strength is presented based on the experimental data and the numerical simulation method. The influence of different environments on steel fiber-reinforced concrete is analyzed, and the grey correlation degree of axial compressive strength is analyzed. The experimental results show that steel fiber can effectively improve the concrete axial bearing capacity, but different responses are observed under the various erosion conditions. A freeze-thaw cycle environment has the most significant impact on the axial compressive strength of concrete, followed by the sulfuric acid environment, and other environments have a weaker impact. The research results will provide a theoretical basis for predicting the performance deterioration of SFRC concerning other erosion conditions and periods.

## 1. Introduction

Concrete is a composite material with high overall strength, excellent durability, good workability, and volume stability. It is widely used in ocean dam waterproofing, airport pavements, long-span steel bridge deck pavement, railway, and national defense [1,2,3,4,5,6,7]. However, concrete is easily corroded due to its characteristics and the influence of the surrounding environment, and its durability is insufficient.

The service life of concrete buildings and structures is closely related to the durability of the materials. Due to the differences in use function and environment, concrete materials are often corroded by various ions, which significantly impacts the durability of concrete structures [8,9,10]. In order to improve the performance of concrete, many new concrete materials have appeared one after another [11,12,13], and fiber concrete is one of them. Adding fiber to concrete can improve the concrete’s strength, toughness, and durability. As a high-performance material, fiber concrete has been widely used in civil engineering construction [14]. Adding fiber to concrete can improve concrete’s mechanical properties and durability [15,16,17,18]. Steel fiber or hybrid steel fiber concrete is the most widely used fiber concrete [19,20,21,22]. However, these studies mainly focus on the mix ratio of steel fiber concrete or steel fiber concrete in natural conditions or the mechanical properties in natural conditions. Frost resistance and fatigue resistance are essential technical indicators of the durability of high-performance concrete. Scholars at home and abroad have researched the deterioration and mechanical properties of high-performance concrete caused by salt-freezing erosion and freezing-thawing damage [23,24,25,26]. The results show that adding fiber is one of the effective ways to improve the strength and toughness of high-performance concrete. The lower the air humidity and the higher the salt solution concentration, the higher the water evaporation rate [27,28]. Scholars have carried out extensive research on the performance of concrete under the coupling action of load and salt erosion, prediction of stress and service life of road concrete under the coupling action of fatigue load and sulfate erosion, and put forward a method of reducing the water–cement ratio and coating silane to improve the sulfate erosion resistance of concrete [29,30,31]. The fatigue damage of concrete in a magnesium sulfate corrosion environment is analyzed. The leading causes of concrete corrosion fatigue damage are microcracks caused by expansion products and microcracks caused by fatigue loads [32]. It has been proven that concrete corrosion and fatigue damage promote each other under the coupling action [33]. Most studies focus on ordinary concrete but do not consider the broad application of fiber concrete in airports. Due to the addition of fiber, the performance of concrete has changed a lot. However, the current research is only limited to the influence of fiber on the mechanical properties of concrete [34,35] or sulfate attack resistance [36,37]. There are few studies on the performance of fiber-dimensional concrete under the coupling of load and sulfate attack. The study selected the number of dry-wet cycles of sulfate, fatigue loads, and fiber content as the influencing factors of the three-factor and three-level experiment [38].

The damage mentioned above and degradation of concrete in a single environment are the main ones, but a comprehensive comparative study on microstructure and axial bearing capacity of steel fiber reinforced concrete under freeze-thaw cycles and various chemical attacks are lacking. Therefore, a durability test was carried out to study the mechanical properties of steel fiber reinforced concrete (SFRC) with different content in different erosion environments. In this study, three kinds of steel fiber content were designed, and steel fiber-reinforced concrete specimens with volume fractions of 0%, 1%, and 2% were made for comparative study. At the same time, six external environments, including water, sodium chloride solution, sodium sulfate solution, dilute sulfuric acid solution, sodium hydroxide solution, and a freeze-thaw cycle were set up. Furthermore, six cylindrical specimens (4 in each group) were made for the axial compression and cross-section microscopic observation tests. By comparing the compressive strength and microstructure of SFRC with different dosages in different erosion environments, the correlation between SFRC damage and environment is analyzed, which provides a theoretical basis for durability design and evaluation of SFRC structure and ensures its safety in structural engineering.

## 2. Materials and Methods

### 2.1. Specimens Preparation

The specimen components include coarse aggregate, fine aggregate, cement, water, and steel fiber. Gravel is used as coarse aggregate with a silt content of 0.7%, divided into two particle size zones: 4.75 mm to 9.5 mm and 9.5 mm to 19 mm, with a mixing ratio of 3:7. Natural river sand is used as fine aggregate with a water content of 4% and a fineness modulus of 2.3. All indexes meet the specification requirements, as shown in Figure 1. P.O42.5 ordinary Portland cement is selected, with a density of 3160 kg/m and a 28-day flexural strength of 8.2 MPa. Water is city tap water.

The steel fiber was milled from steel ingots produced by Shanghai Helex Technology Co., Ltd. (Shanghai, China) is shown in Figure 2, and its specification information is shown in Table 1.

The compressive strength grade of concrete is C30, and a water–cement ratio of 0.54 is used [39]. Therefore, only the change in steel fiber content is considered in this test, and the effect of water consumption is ignored. Table 2 shows the concrete mix design.

In this experiment, the volume content of steel fiber is selected for 0%, 1%, and 2%, respectively. The size of concrete compressive strength specimens is 150 × 150 × 150 mm^3^ cube, with three specimens in each group and nine specimens with three different steel fiber contents. The axial compression concrete test specimens used Φ100 × 200 mm^3^ cylinders. There are four samples in each group (including 3 for mechanical and 1 for microscopic observations). Three kinds of concrete cylindrical specimens with different steel fiber contents and six different corrosive environments (water, sodium chloride solution, sodium sulfate solution, dilute sulfuric acid solution, sodium hydroxide solution, and freeze-thaw cycle) were selected, and 156 specimens were used in the experiment.

Coarse aggregate and fine aggregate were sieved and weighed, then put into a concrete mixer for dry mixing for 2 min, then water was added for mixing for 3–5 min, and finally evenly sprinkled with a specified amount of steel fiber for mixing for 2–5 min. Next, coarse and fine aggregate were sieved, weighed and put in a concrete mixer for dry mixing for 2 min, then water was added and stirred for 3–5 min. Finally, a specified amount of steel fiber was evenly sprinkle and stirred for 2–5 min. Next, the prepared fiber concrete was added into the mold twice, and vibrated for 20–30 s. During the vibrating process, the vibrating rod is inserted and tamped many times to improve the compactness. After curing for 24 h, the mold was removed and put into the curing room for curing for 28 days, as shown in Figure 3.

### 2.2. Experimental Method

Six environmental factors were selected in this study: water, sodium chloride solution, sodium sulfate solution, dilute sulfuric acid solution, sodium hydroxide solution, and a freeze-thaw cycle. The specific test method is as follows.

The laboratory rapid freeze-thaw method and KRD-V5 concrete rapid freeze-thaw testing machine were used. At the end of each freezing cycle, the center temperature of the sample was −18 °C ± 2 °C. Similarly, at the end of the melting cycle, the center temperature of the sample is +7 °C ± 2 °C, and a freeze-thaw cycle is completed in 2–4 h. The specimen was placed in a freeze-thaw tester, water added, and the liquid level was 5 cm above the surface of the specimen. The number of large cycles was set to 100 and minor cycles to 25 times.

The 3.5%NaCl solution, 10%Na_2_SO_4_ solution, 5%H_2_SO_4_ solution and 2 mol/L NaOH solution were prepared as shown in Figure 4, the prepared 30 L solution was poured into a large storage box, and 30 L water was put into the same large storage box as the control group.

The cured cylindrical specimen was placed as shown in Figure 5, with the liquid level 15 cm away from the top surface of the specimen. The lid was covered to avoid water evaporation.

A digital optical microscope was used (Figure 4) to observe the microstructure, the specimen surface (position 1), cross-section edge (position 2), and cross-section center (position 3) are observed, as shown in Figure 5. The digital optical microscope used in this experiment is GP-500, produced by Kunshan Gaopin Precision Instrument Co., Ltd. (Kunshan, China), with a magnification of 1–500 times. Therefore, the magnification chosen in this study is 100 times. The lighting range is 0–30,000 LUX, which is adjustable by wire. The lighting mode is eight white LED lamps, the video screen specification is AVI, and the lens speed is 30 F/s.

The bearing capacity under axial pressure was tested (Figure 6). When subjected to the freeze-thaw cycle and chemical immersion, all samples were continuously and evenly loaded until failure, and the loading speed was 0.5 MPa/s.

## 3. Results and Discussion

### 3.1. Microscopic Morphological Analysis

After 100 freeze-thaw cycles, soaking in sodium chloride solution, sodium sulfate solution, sulfuric acid solution, and sodium hydroxide solution for 28 days. The surface (position 1) and cross-section (position 2, position 3) of three kinds of specimens with different fiber contents were observed microscopically, and 100 times magnification was selected to compare the influence degree of different environments on the specimens.

Figure 7 shows position 1 after six kinds of environmental erosion when the steel fiber content is 0%. The microscopic observation results show that the surface crack of the specimen in a freeze-thaw cycle environment is the largest when the fiber is not added. However, there is no crack on the surface after water immersion, and the degree of damage to the specimen in water immersion is the smallest. All the specimens in other environments were damaged, but the degree is less than the freeze-thaw cycle.

Figure 8 shows position 1 after six kinds of environmental erosion when the steel fiber content is 1%. After adding 1% steel fiber into the specimen, compared with the six environments, there are more cracks in the microscopic pictures of the freeze-thaw cycle and sulfuric acid immersion, which have the most significant impact on the specimen. No cracks appear after water immersion; cracks appear after immersion in other environments, but the number of cracks is small.

Figure 9 shows position 1 after six kinds of environmental erosion when the steel fiber content is 2%. After adding 2% steel fiber into the specimen, it can be concluded that the freeze-thaw cycle and sulfuric acid environment have more apparent damage to the specimen, followed by other environments, and the damage after water immersion is the least obvious.

After different kinds of environmental erosion, except water immersion, cracks appeared on the surfaces of different kinds of micro-concrete specimens under the action of other environments. From the microscopic results, comparing the number and depth of cracks, adding steel fiber did not inhibit the cracks on the surfaces of specimens. Compared with six kinds of erosion environments, the freeze-thaw cycle and sulfuric acid environment have the most decisive influence on the surface cracks for the three kinds of specimens. In contrast, water has a minor influence on the cracks.

Microscopic observation of position 2 after environmental erosion shows that the magnification is 100 times. Figure 10 shows the observation results of six kinds of environmental erosion when the steel fiber content is 0%. Cracks appear on the specimen’s cross-section after the freezing-thawing cycle and NaCl environment erosion. However, no cracks appear after other environmental erosion, which indicates that the freezing-thawing cycle and NaCl environment greatly influence concrete damage, while other environments have little influence.

Microscopic observation of position 2 after environmental erosion shows that the magnification is 100 times. Figure 11 shows the observation results of six kinds of environmental erosion when the steel fiber content is 1%. After adding 1% steel fiber, only the specimen’s cross-section is cracked after the freeze-thaw cycle environment erosion. However, no cracks are found after other environments’ erosion, which shows that the freeze-thaw cycle damage to the specimen is more severe than in other environments.

Microscopic observation of position 2 after environmental erosion shows that the magnification is 100 times. Figure 12 shows the observation results of six kinds of environmental erosion when the steel fiber content is 2%. The microscopic results of the specimen’s cross-section are similar to those of the specimen with 1% steel fiber content, except that the crack occurs at position 2 of the specimen’s cross-section after the freeze-thaw cycle. However, the difference is that the crack of the concrete specimen with 2% steel fiber content is minor.

Comparing the microscopic observation results of position 2, the concrete without steel fiber is cracked after the freeze-thaw cycle and NaCl environmental erosion. However, only the freeze-thaw cycle environmental erosion causes cracks in the specimen after adding steel fiber. Comparing the two dosages, the concrete cracks with 2% content are minor, which shows that steel fiber can inhibit the internal cracks of concrete, thus reducing the environmental damage to concrete. Adding 2% concrete is more effective than 1% concrete. It can be known that the freeze-thaw cycle has the most significant influence on the damage to concrete specimens in the six erosion environments. Comparing the microscopic results of position 3, as shown in Figure 13, cracks are produced in specimens with different fiber contents after the freeze-thaw cycle environment erosion. The width and length of cracks decrease with the increase in steel fiber content.

There is no crack in the center of the concrete cross-section with different fiber content in the other five environments. In this paper, after soaking in NaCl solution, as shown in Figure 14, no crack is found in the specimens with each of the three fiber contents.

Comparing the three locations, except for the water environment, all the specimens with three dosages have cracks on their surfaces in location 1. Similarly, except for the cracks caused by the freeze-thaw cycle, only the cracks were caused by plain concrete after soaking in NaCl solution in location 2. Therefore, the environmental erosion on the specimen surface is more significant than inside, indicating environmental damage is from outside to inside. Compared with positions 2 and 3, the same point is that freeze-thaw cycles have destroyed concrete specimens under six kinds of environmental erosion. The length and width of cracks decrease with the increase of steel fiber. This observation shows that the freeze-thaw cycle has the most significant influence on the damage to concrete, and adding steel fiber can inhibit the development of cracks in concrete specimens. That is, it can inhibit the damage to concrete.

### 3.2. Analysis of Mechanical Deterioration Performance of SFRC

The concrete cubic compressive test is set up in Figure 15, and the test results are shown in Table 3.

The test shows that the increase in steel fiber content has a positive effect on the compressive strength of SFRC. When the fiber content is 1% and 2%, the compressive strength increases by 11.4% and 30.6%, respectively. The addition of steel fiber can inhibit the transverse deformation of concrete and improve the compressive strength of concrete. However, due to the existence of steel fibers, a harmful interface is formed between the concrete and steel fiber. The interface area is suddenly destroyed when the compression load reaches a specific value.

Cylindrical concrete specimens with a steel fiber content of 0%, 1%, and 2% were subjected to an axial compression test after 100 freeze-thaw cycles and 28 days in different solutions. An axial compression test is carried out on cylinder specimens before erosion, and the test results are shown in Table 4.

A schematic diagram of the axial compression test of specimens with other volume content of steel fiber is shown in Figure 16. The more steel fiber content, the less noticeable the surface failure of the specimen, and the better the specimen’s integrity after failure, the less the surface mortar falls off.

The axial compression test results of cylindrical specimens are shown in Table 5.

The table can show that the intensity changes are different under different erosion environments. After adding steel fiber into concrete, the compressive strength of concrete in various environments mainly increases with the increase in steel fiber. Compared with ordinary concrete, the compressive load of the steel-fiber concrete is improved. When the pressure direction of fiber in the member is oriented vertically, it can effectively stop the transverse deformation and show the best reinforcement effect. With the increase in soaking time, the soaked water specimens are considered the control group. The axial compressive strength of the specimens soaked in other solutions decreased, indicating that various adverse environments have large or small destructive effects on the concrete specimens. The steel fiber can effectively prevent the corrosion of the chemical environment on the concrete. Generally speaking, the more steel fiber is added, the higher the strength of the specimen. However, with the decrease in erosion failure strength, the strength of the specimen with the same fiber volume content is different in different erosion environments. The influence of different environments on the strength of steel fiber-reinforced concrete needs further study.

The experimental data show that adding steel fiber in different erosion environments is beneficial. Adding more steel fiber in most environments will have better mechanical properties, effectively resisting the damage of adverse environments. However, in the NaCl and Na_2_SO_4_ environment, adding 2% steel fiber is worse than adding 1% steel fiber, and it also reduces the mechanical properties of concrete. After adding 2% steel fiber, the mechanical properties of specimens are less than those of plain concrete, which makes the addition of fiber negatively correlated with compressive strength, so it is not always the case that adding more steel fiber will achieve better results. Therefore, it is essential to carry out experimental research before steel fiber is applied to practical construction projects, which can avoid the waste of material resources and the adverse effects caused by improper mix proportions.

Adding excessive steel fiber in the environment of sodium chloride and sodium sulfate not only causes material waste, but also adversely affects the performance of building structures.

According to the microscopic observation results, it can be seen that freeze-thaw cycles have the greatest impact on the concrete specimen. According to the strength changes before and after 100 freeze-thaw cycles, the strength loss rate is calculated, which is the ratio of the change before and after erosion to the compressive strength before erosion. The strength increase rate is the ratio of the strength after erosion to the compressive strength before erosion of the plain concrete. After 100 freeze-thaw cycles, the compressive strength of the specimens with 0%, 1% and 2% content lost 35.0%, 33.7% and 34.5% respectively. The compressive strength of steel fiber reinforced concrete specimens with 1% and 2% content increased by 17.6% and 69.2% respectively. From the calculated value, it can be concluded that adding steel fiber can effectively inhibit the damage of concrete caused by freeze-thaw cycle, and the effect of adding 2% steel fiber is better than that of adding 1% volume fraction steel fiber.

### 3.3. Data Fitting Analysis

After freezing and thawing for 100 times and soaking for 28 days, the relationship between all data of axial compressive strength of various cylindrical specimens and steel fiber content was numerically simulated and analyzed. The fitting curve is shown in Figure 17.

The figure shows that concrete with different fiber contents has different erosion degrees in different environments. After adding steel fiber, the axial compressive strength of concrete is improved, and the optimum range of steel fiber is different in different environments. The regression analysis results are shown in Table 6.

The results show that concrete with different fiber content has different erosion degrees in different environments. After adding steel fiber, the axial compressive strength of concrete is improved, and the optimum range of steel fiber is different in different environments. The addition of steel fiber significantly improves the compressive strength of concrete, but in not all environments, the more steel fiber is added, the better. Adding too much steel fiber in a sodium chloride environment and sodium sulfate environment will be counterproductive. According to the test results, the optimal dosage is 1% in a sodium sulfate and sodium chloride environment, and 2% in a sulfuric acid and sodium hydroxide environment.

### 3.4. Grey Correlation Analysis

Grey relational analysis is a statistical analysis technique that originates from the grey system theory in the system science theory. It is suitable for the quantitative analysis of the dynamic development process, and there will be no discrepancy between the quantitative and qualitative results.

Compared with traditional analysis methods, such as variance analysis or regression analysis, which only take a single index as the measurement standard, the grey correlation degree analysis can comprehensively and genuinely reflect the changes of comprehensive factors by comparing multiple traits. As a result, its comprehensive evaluation results are more scientific and accurate [40,41,42].

Grey correlation analysis theory is an integral part of grey system theory, and correlation is the correlation between the reference and influencing factors. The greater the correlation, the greater the correlation value, and the more sensitive the influencing factors are to comparative factors. This study divides the grey correlation analysis into six steps [43,44].

Step 1: Select the reference sequence X_0_ = (x_01_, x_02_, x_03_, x_04_, x_05_, x_06_, x_07_), Compare the sequence Xi = (x_i1_, x_i2_, x_i3_, x_i4_, x_i5_, x_i6_, x_i7_), where i = 1, 2, 3…, n. The data sequence that reflects the characteristics of system behavior is called the reference sequence. A data sequence composed of factors that affect system behavior is called a comparative sequence.

This paper takes the axial compressive strength of concrete specimens soaked in water as the reference sequence X_0_. The axial compressive data sequences of concrete specimens soaked in sodium chloride solution, sodium sulfate solution, dilute sulfuric acid solution, and sodium hydroxide solution and the frozen-thawed cycle is taken as the comparison sequences. From the numerical simulation analysis, it can be seen that the optimal content of steel fiber is different in different environments. However, considering the economy and applicability, the data of 1% volume content of steel fiber is selected in the grey correlation analysis, and the data of reference sequence and comparison sequence are shown in Table 7.

Step 2: Dimensionless processing of variables. Due to the different physical meanings of the factors in the system, the data dimensions are not necessarily the same, which makes it inconvenient to compare or difficult to obtain a correct conclusion when comparing. Therefore, dimensionless data processing is generally required when analyzing the grey correlation degree. Common methods include the initial value method and the average value method. Here, the initial value method is used to obtain X’_i_ = X_i_/xi1 = (x’i1, x’i2, …, x’in), i = 0, 1, 2, 3…, m. The above Table 5 is dimensionless by the initial value method, and Table 8 is obtained.

Step 3: Find the business trip sequence, maximum and minimum difference.

Sequence difference is Δ_0i_(k) = |x’_0_(k) − x’_i_(k)|, k = 1, 2, …, n.

The maximum difference of n is: M = Maxi Maxk Δi (k), and the minimum difference is: m = Mini Mink Δi(k).

The difference sequence values are calculated from Table 6, and as shown in Table 9, the difference sequence values are Δ01 = (0, 0.057, 0.586), Δ02 = (0, 0.624, 0.216), Δ03 = (0, 0.068, 0.168), Δ04 = (0, 0.075, 0.219) and Δ05 = (0, 0.039, 0.172)

Step 4: Calculate the correlation coefficient. The degree of correlation is essentially the difference in geometric shapes between curves. Therefore, the difference between curves can be used to measure the correlation degree.

r(x0(k), xi(k)) = (m + §M)/(Δ0i(k) + §M), § ∈ (0, 1). k = 1, 2, …, n. i = 0, 1, 2, …, m. Where § is the resolution coefficient, which is often taken as 0.5.

The maximum difference between two poles M = 0.624, the minimum difference between two poles m = 0, and the resolution = 0.5, respectively, can be used to obtain the grey correlation coefficient and grey correlation degree (see Table 10).

Step 5: Find the correlation degree. Because the correlation coefficient is the correlation degree value between the comparison series and the reference series at each moment (each point in the curve), it has more than one number. The information is too scattered to make an overall comparison. Therefore, it is necessary to concentrate the correlation coefficient of each moment (each point in the curve) into one value, that is, to find its average value, which can be used as a quantitative expression of the correlation degree between the comparison series and the reference series.


$$r(x_{0},\;x_{i})=\frac{1}{n}\sum_{ki=1}^{n}r\left(x_{0}\left(\;k\right)\;,\;x_{i}\left(\;k\right)\;\right).\;i=0,\;1,\;2,\;\ldots ,\;m$$


According to the above results, we can know that r(x_0_, x_1_) = 0.567, r(x_0_, x_2_) = 0.642, r(x_0_, x_3_) = 0.824, r(x_0_, x_4_) = 0.798, r(x_0_, x_5_) = 0.845.

Step 6: Analyze the results. If r(x_0_, x_i_) > r(x_0_, x_j_) > r(x_0_, x_k_) > … > r(x0, xz), it means that xi is better than x_j_, xj is better than x_k_, and so on. x_i_ > x_j_ > x_k_ > … > x_z_. x_i_ > x_j_ indicates that the grey correlation degree of factor x_i_ to reference sequence x_0_ is greater than x_j_. The greater the correlation, the stronger the closeness between this group of factors and parent factors.

The influence of environment on the axial compressive strength of steel fiber-reinforced concrete can be explained by the coefficient from large to small, followed by freeze-thaw cycles, dilute sulfuric acid, sodium hydroxide, sodium sulfate and sodium chloride.

This shows that the freeze-thaw cycle environment has the closest relationship with the axial compressive strength of the specimen. It has the most significant influence on the mechanical properties of the specimen, followed by sulfuric acid environment, sodium hydroxide, sodium sulfate environment, and sodium chloride environment. The gray correlation coefficient between the axial compressive strength of steel fiber-reinforced concrete and freeze-thaw cycle erosion is the largest. It is the most severely damaged by freeze-thaw. Concrete’s overall axial compressive strength is improved after adding steel fiber, so we can continue to study adding fiber materials to concrete or improving the concrete formula to weaken the impact of this erosion.

Similarly, the sulfuric acid environment significantly impacts concrete, and its correlation is slightly less than that of the freeze-thaw cycle. However, it still plays an essential role in concrete damage. Therefore, to avoid this kind of damage in the actual design, exposure time to a sulfuric acid environment needs to be reduced.

The results of grey correlation analysis verify the microscopic observation results, and the freeze-thaw cycle has the most significant influence on the failure of concrete specimens.

## 4. Conclusions

In this paper, two variables of steel fiber content and erosion environment are designed, and microscopic observation, cube compression test, and axial compression test are carried out on the specimens. As a result, the following conclusions can be obtained.

Steel fiber can effectively inhibit the erosion of concrete in an unfavorable environment. In addition, the incorporation of steel fiber can inhibit the mortar from falling off on the concrete surface and reduce cracks on the concrete surface, thus improving the frost resistance of concrete.It can be seen from microstructure observation that the external environment damages concrete from the outside to the inside. With the increase of erosion time, concrete cracks are developed, and adding steel fiber can inhibit cracks and improve the durability of concrete.In the freeze-thaw environment, the freeze-thaw cycle makes the compressive strength of concrete lose 33.7–35%. However, adding steel fiber can increase the compressive strength of concrete by 17.6–69.2%, thus improving the frost resistance of concrete. Therefore, it is beneficial to add steel fiber to concrete in the freeze-thaw environment.According to the axial compressive test results of concrete in six erosion environments, it can be seen that different erosion environments have different influences on the axial compressive strength of concrete. The analysis and calculation of the compressive strength according to the grey correlation degree can be concluded that the damage degree of the unfavorable environment to concrete is freezing and thawing cycle > dilute sulfuric acid > sodium hydroxide > sodium sulfate > sodium chloride.The best content of steel fiber is different in various erosion environments (water, sodium chloride solution, sodium sulfate solution, dilute sulfuric acid solution, sodium hydroxide solution, and freeze-thaw cycle). For example, the best content is 2% in the freeze-thaw cycle, dilute sulfuric acid and sodium hydroxide, and the best content is 1% in sodium chloride and sodium sulfate. Adding excessive steel fiber in the environment of sodium chloride and sodium sulfate not only causes material waste, but also adversely affects the performance of building structures.

This study only compares the existence, length, and quantity of cracks in different erosion environments and cannot find out the influence of different environments on the development of concrete surface and internal cracks. Furthermore, there is limited research on crack width and specific crack size, which is also the limitation and deficiency of this research, and further research is necessary.

## Figures and Tables

**Figure 1 materials-15-04748-f001:**
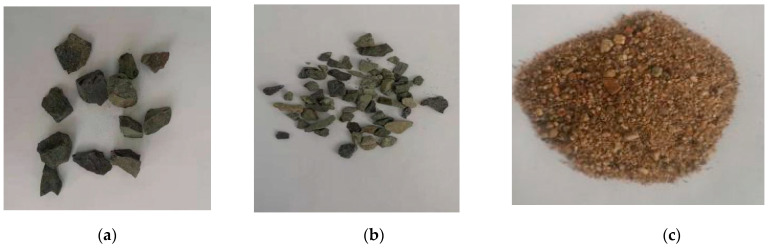
Particle size distribution. (**a**) Particle size 4.75–9.5 mm. (**b**) Particle size 9.5 mm–19 mm. (**c**) Medium sand.

**Figure 2 materials-15-04748-f002:**
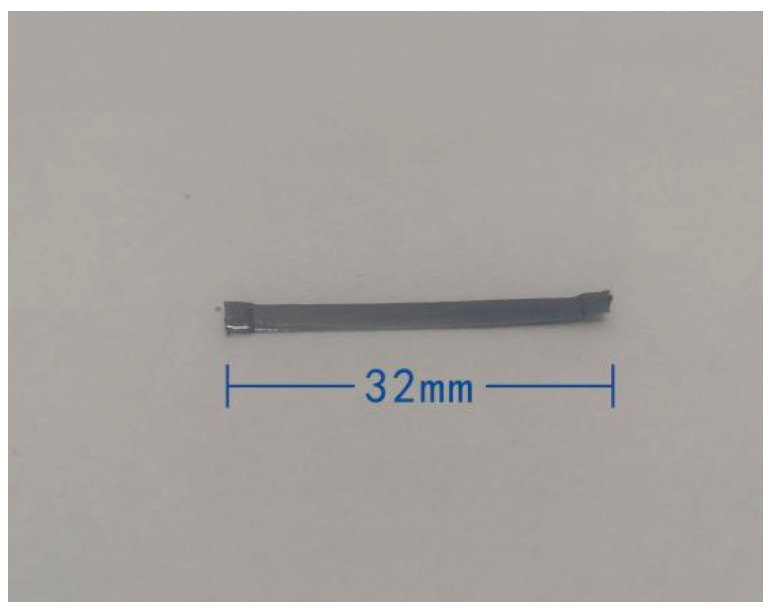
Steel fiber.

**Figure 3 materials-15-04748-f003:**
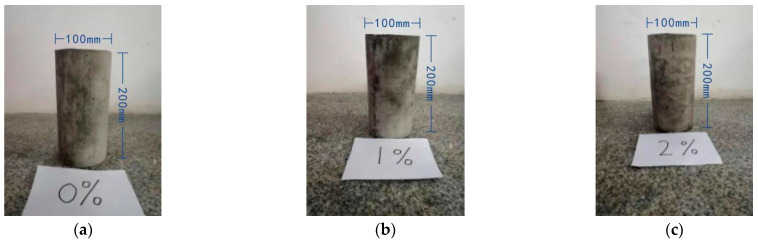
SFRC concrete cylinders. (**a**) Steel fiber content 0%. (**b**) Steel fiber content 1%. (**c**) Steel fiber content 2%.

**Figure 4 materials-15-04748-f004:**
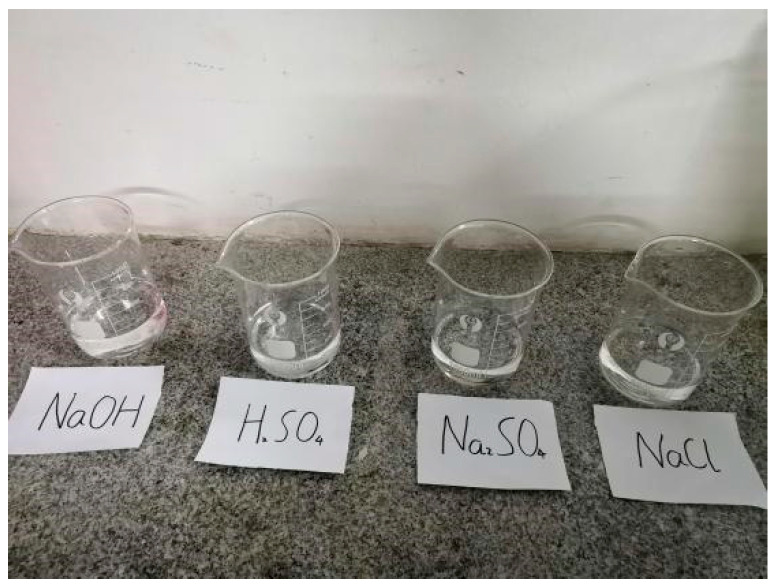
Chemical solution.

**Figure 5 materials-15-04748-f005:**
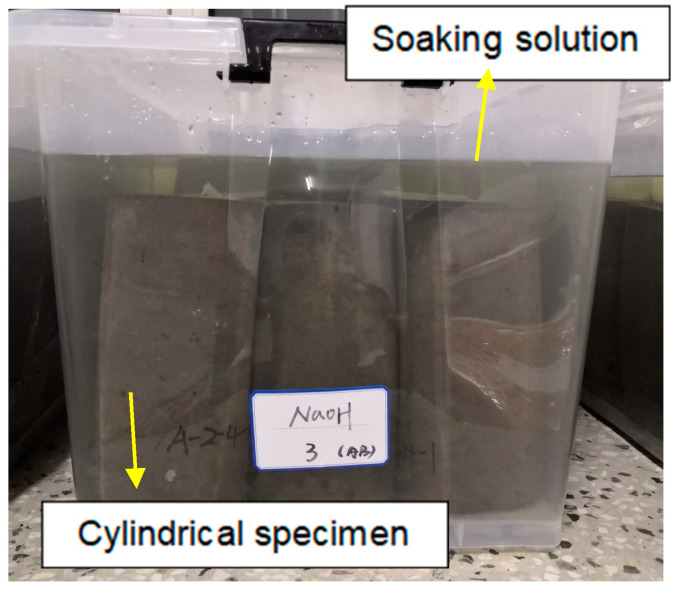
Test specimens immersed in storage box.

**Figure 6 materials-15-04748-f006:**
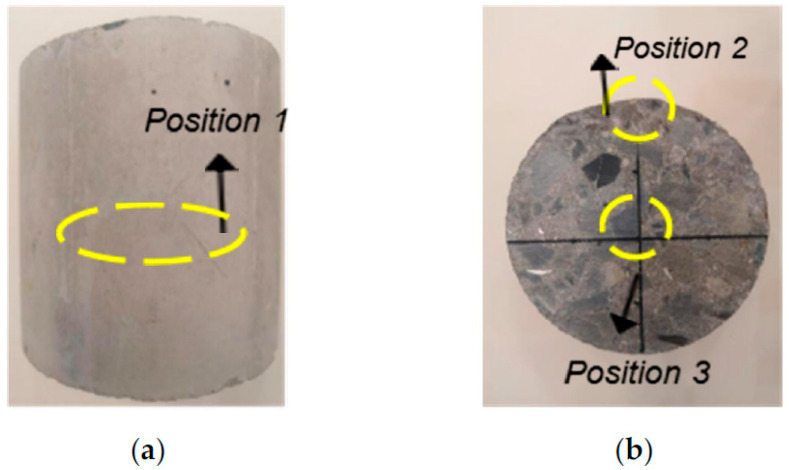
Microscope observation position. (**a**) Test piece surface. (**b**) Cross section of specimen.

**Figure 7 materials-15-04748-f007:**
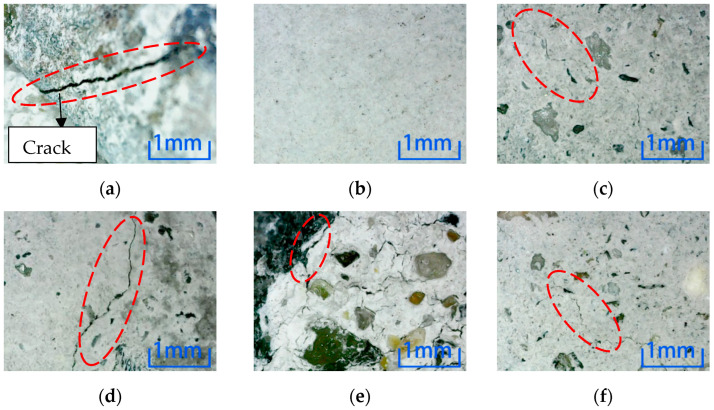
Microscopic observation of 0% content specimen at position 1. (**a**) Freeze-thaw cycle. (**b**) Water. (**c**) NaCl solution. (**d**) Na_2_SO_4_ solution. (**e**) H_2_SO_4_ solution. (**f**) NaOH solution.

**Figure 8 materials-15-04748-f008:**
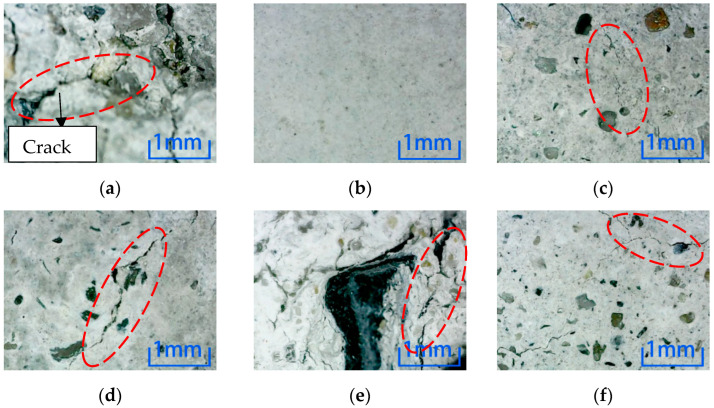
Microscopic observation of 1% content specimen at position 1. (**a**) Freeze-thaw cycle. (**b**) Water. (**c**) NaCl solution. (**d**) Na_2_SO_4_ solution. (**e**) H_2_SO_4_ solution. (**f**) NaOH solution.

**Figure 9 materials-15-04748-f009:**
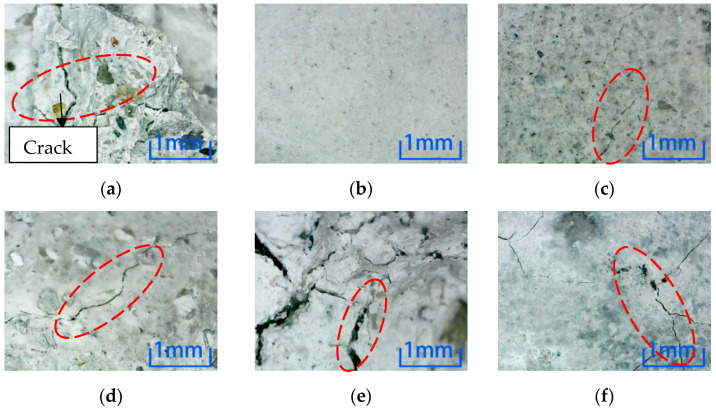
Microscopic observation of 2% content specimen at position 1. (**a**) Freeze-thaw cycle. (**b**) Water. (**c**) NaCl solution. (**d**) Na_2_SO_4_ solution. (**e**) H_2_SO_4_ solution. (**f**) NaOH solution.

**Figure 10 materials-15-04748-f010:**
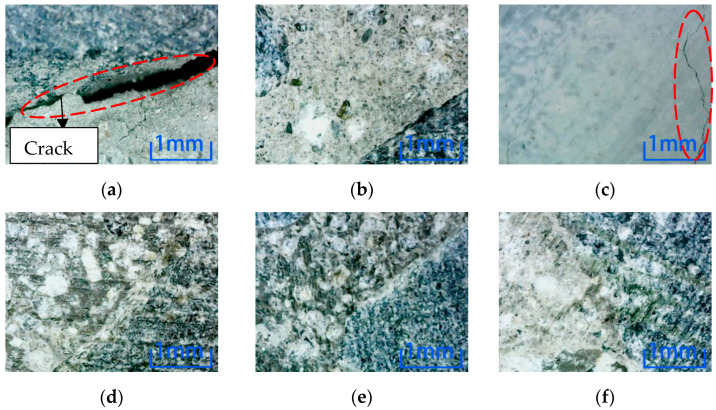
Microscopic observation of 0% content specimen at position 2. (**a**) Freeze-thaw cycle. (**b**) Water. (**c**) NaCl solution. (**d**) Na_2_SO_4_ solution. (**e**) H_2_SO_4_ solution. (**f**) NaOH solution.

**Figure 11 materials-15-04748-f011:**
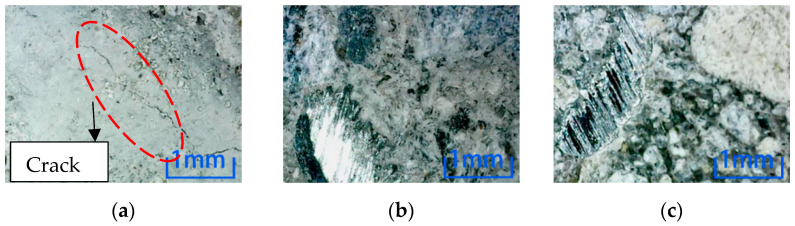
Microscopic observation of 1% content specimen at position 2. (**a**) Freeze-thaw cycle. (**b**) Water. (**c**) NaCl solution. (**d**) Na_2_SO_4_ solution. (**e**) H_2_SO_4_ solution. (**f**) NaOH solution.

**Figure 12 materials-15-04748-f012:**
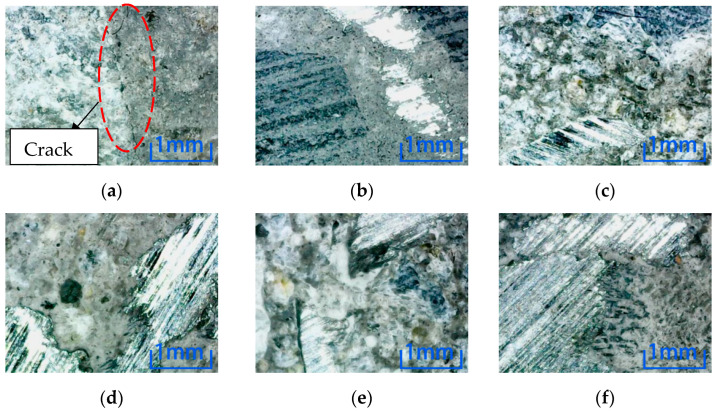
Microscopic observation of 2% content specimen at position 2. (**a**) Freeze-thaw cycle. (**b**) Water. (**c**) NaCl solution. (**d**) Na_2_SO_4_ solution. (**e**) H_2_SO_4_ solution. (**f**) NaOH solution.

**Figure 13 materials-15-04748-f013:**
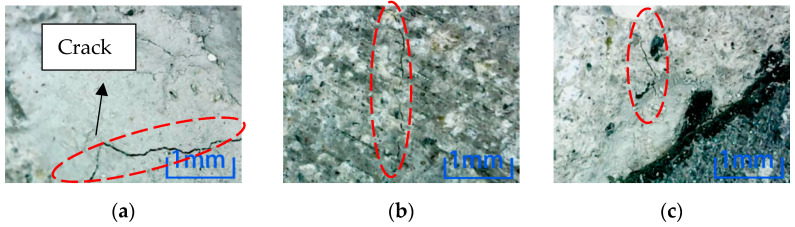
Microscopic observation of position 3 in 100 times freeze-thaw cycles. (**a**) Steel fiber content 0%. (**b**) Steel fiber content 1%. (**c**) Steel fiber content 2%.

**Figure 14 materials-15-04748-f014:**
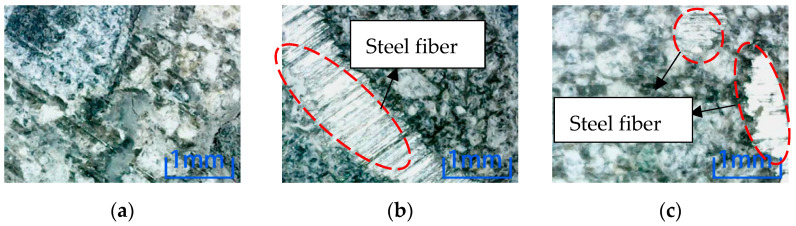
Microscopic observation of specimen soaked in NaCl solution at position 3. (**a**)Steel fiber content 0%. (**b**) Steel fiber content 1%. (**c**) Steel fiber content 2%.

**Figure 15 materials-15-04748-f015:**

Concrete cubic compressive test. (**a**) Steel fiber content 0%. (**b**) Steel fiber content 1%. (**c**) Steel fiber content 2%.

**Figure 16 materials-15-04748-f016:**
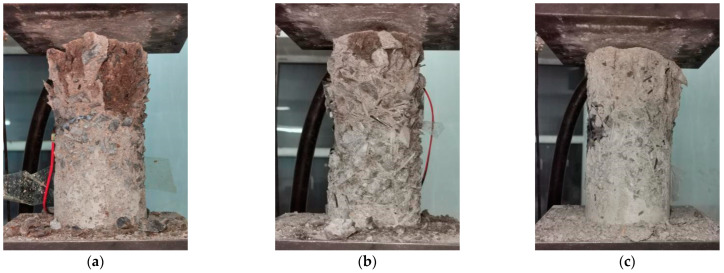
Axial compression test. (**a**) Steel fiber content 0%. (**b**) Steel fiber content 1%. (**c**) Steel fiber content 2%.

**Figure 17 materials-15-04748-f017:**
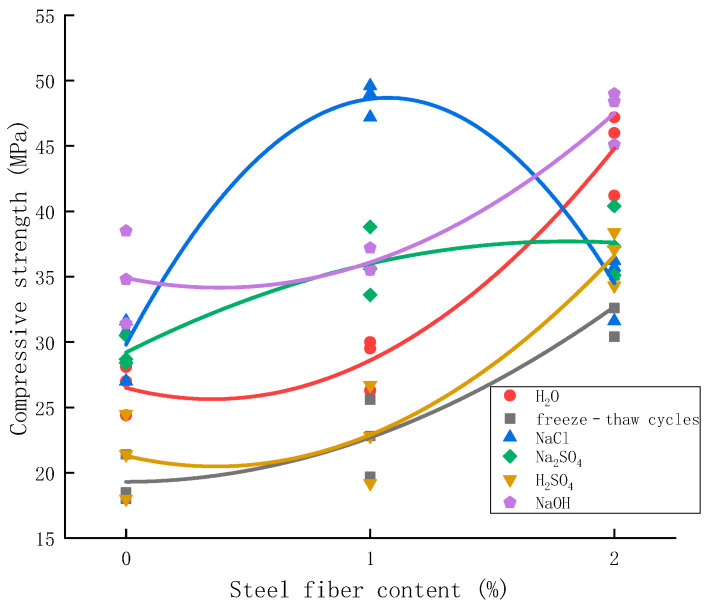
Regression analysis fitting curve.

**Table 1 materials-15-04748-t001:** Specification of steel fiber.

Tensile Strength	Length	Width	Draw Ratio	Density
700 MPa	32 mm	2.6 mm	40	7850 kg/m³

**Table 2 materials-15-04748-t002:** The mix design of steel fiber concrete (kg/m³).

Material	Coarse Aggregate	Sand	Cement	Water	Steel Fiber
Plain concrete	1169	635	387	209	0
1% steel fiber-reinforced concrete	1169	635	387	209	78.5
2% steel fiber-reinforced concrete	1169	635	387	209	157

**Table 3 materials-15-04748-t003:** Compressive strength of concrete cube.

Type (Steel Fiber Content)	0%	1%	2%
Compressive strength (MPa)	30.7 MPa	34.2 MPa	40.1 MPa

**Table 4 materials-15-04748-t004:** Axial compressive strength of concrete.

Type (Steel Fiber Content)	0%	1%	2%
Compressive strength (MPa)	30.1 MPa	34 MPa	49.8 MPa

**Table 5 materials-15-04748-t005:** Axial compressive strength of concrete (MPa).

Environmental Erosion Conditions	1% (14d)	2% (14d)	1% (28d)	2% (28d)
H_2_O	34.3 MPa	49.8 MPa	28.6 MPa	44.8 MPa
Freeze-thaw cycles	33.0 MPa	40.9 MPa	22.7 MPa	32.7 MPa
NaCl	53.8 MPa	38.1 MPa	48.6 MPa	34.5 MPa
Na_2_SO_4_	55.6 MPa	36.5 MPa	35.9 MPa	37.4 MPa
H_2_SO_4_	32.0 MPa	39.8 MPa	22.8 MPa	36.6 MPa
NaOH	36.9 MPa	48.4 MPa	36.1 MPa	47.7 MPa

**Table 6 materials-15-04748-t006:** Results of concrete strength analysis.

Environmental Erosion Conditions	Regression Equation	R^2^	RSS
H_2_O	fc = 7.05x^2^ − 4.95x + 26.5	0.94	35.44
Freeze-thaw cycles	fc = 3.3x^2^ + 0.1x + 19.3	0.89	35.22
NaCl	fc = −16.45x^2^ + 35.25x + 29.8	0.95	27.94
Na_2_SO_4_	fc = −2.6x^2^ + 9.4x + 29.2	0.80	30.52
H_2_SO_4_	fc = 6.05x^2^ − 4.45x + 21.3	0.88	58.06
NaOH	fc = 5.1x^2^ − 3.9x + 34.9	0.89	35.86

**Table 7 materials-15-04748-t007:** Optimum content of steel fiber in different erosion environments.

Soak Period	H_2_O (X_0_)	NaCl (X_1_)	Na_2_SO_4_ (X_2_)	H_2_SO_4_ (X_3_)	NaOH (X_4_)	Freeze-Thaw Cycles (X_5_)
0	34.20	34.20	34.20	34.20	34.20	34.20
14	34.33	53.83	55.68	32.00	36.90	33.00
28	28.60	48.63	35.98	22.85	36.10	22.71

**Table 8 materials-15-04748-t008:** Dimensionless processing table.

Soak Period	H_2_O (X_0_)	NaCl (X_1_)	Na_2_SO_4_ (X_2_)	H_2_SO_4_ (X_3_)	NaOH (X_4_)	Freeze-Thaw Cycles (X_5_)
0	1	1	1	1	1	1
14	1.004	1.574	1.628	0.936	1.079	0.965
28	0.836	1.422	1.052	0.668	1.056	0.664

**Table 9 materials-15-04748-t009:** Difference sequence table.

Soak Period	NaCl (X_1_)	Na_2_SO_4_ (X_2_)	H_2_SO_4_ (X_3_)	NaOH (X_4_)	Freeze-Thaw Cycles (X_5_)
0(d)	0	0	0	0	0
14(d)	0.570	0.624	0.068	0.075	0.039
28(d)	0.586	0.216	0.168	0.219	0.172

**Table 10 materials-15-04748-t010:** Grey correlation coefficient and grey correlation degree.

Soak Period	r [x0(k), x1(k)]	r [x0(k), x2(k)]	r [x0(k), x3(k)]	r [x0(k), x4(k)]	r [x0(k), x5(k)]
0(d)	1.00	1.00	1.00	1.00	1.00
14(d)	0.354	0.333	0.821	0.806	0.890
28(d)	0.348	0.591	0.650	0.587	0.644
r [x0, xi]	0.567	0.642	0.824	0.798	0.845

## Data Availability

Not applicable.

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
