# Peer review of "Grey Correlation Analysis of the Durability of Steel Fiber-Reinforced Concrete under Environmental Action"

_materials, 2022, doi:10.3390/ma15144748_

Round 1

Reviewer 1 Report

1. 1. please provide professional email address for the corresponding author, no personal email, or without name in it

2. Figure 1 needs a ruler with dimension to understand the scale,

3. Quality of Fig 1 is low

4. C30 grade, relevant standard needed to be cited

5. C30 is cube or cylinder strength?

6. Figure 2, again please add a ruler, dimensions diffiult to interpret

7. Figure 4 could be removed

8. Figure 6 could be removed

9. Table 1, 2, 3 4, add MPa only on the left column

10. reference list is not properly formatted,

11 . In its present form, the paper reads like a lab report. It is strongly recommended that the authors use their test results in combination with some kind of modelling (existing or their own). By doing so the paper would be of greater value to readers of the journal

12.  Limititations of this research are not well addressed and discussed,

13. Coefficient of Variations of results, number of specimens, need to be clearly specified and commented,

14. Conclusions section needs to be revised significantly to reflect the content the manuscript.

15. Grey Correlation Analysis, this part is really unclear and needs to be reformulated and improved, several symbols or acronyms are not very clear

16. Figure 17, add scattering (standard deviation values)

17. Table 4, chemical formulas and elements need revision

18. Mechanical properties of steel fibres are not well discussed and provided, this is really important

Overall, the test results are useful and expand the existing database.  However, the paper has some weaknesses, which make publication in a scientific journal rather questionable.  

Author Response

Dear Reviewer,

The authors would like to thank the reviewers for their review of the manuscript and valuable suggestions. All comments have been carefully considered and accounted for in a new version of the manuscript.

thanks,

Yongcheng Ji

Reviewer 2 Report

The article is interesting.

Indeed, most researchers focus on the mechanical characteristics of fibrobrete. Definitely there are less analyses concerning durability of such concretes. However, in my opinion there are too few references in the article to the literature on the subject (even if only on the mechanical properties). The addition of steel fibers does not always increase the compressive strength of concrete.

The damage assessment is too general and it is difficult to compare the results. It would be better to relate the damage (number of cracks, width of opening) for example to a concrete surface. Has there been any loss of specimen mass? Do fibers always inhibit the development of surface damage? Or is it a random result?

No comparison of concrete compressive strength results before being subjected to destructive actions with results after such destruction. Perhaps use a percentage description of the resulting differences? We generally expect strength decreases as a result of aggressive environments. But how to interpret the increase in strength of concrete after 14 and 28 days of exposure to aggressive environments (e.g. at the beginning the strength of concrete with 1% addition of steel fibers is 34 MPa, and after 14 days in NaCl solution already 53.8 MPa was obtained, to reach 48.6 MPa after 28 days).

Figure 2 is rather redundant. It does not show the different fiber content of the sample. If not for the inscriptions the samples would visually not be different.

Too little commented Figure 17, which brings interesting findings. Line 285 rather incorrectly indicates Table 4.

The conclusions summarize the results of the study. It is unfortunate that there are no recommendations for the amount of steel fiber reinforcement in a given destructive environment.

Author Response

(The authors gave the same response as above.)

Reviewer 3 Report

Dear authors, first of all, please explain and elaborate the novelty of this research. The article is well-organized. Please consider the following comments:

fiber mixing rates (0%, 1%, 2%) by weight of …...? Please specify in abstract and the text.

What is microscopic observation? Did you use SEM, etc. Please find a better word.

Explain the methods related to your capturing system.

Improve the quality of figures, specifically fig. 1.

Add a table for fiber specifications.

Add all the specification for all materials used including cement, aggerates, water etc.

Improve the quality of fig. 17.

Please reformulate the tittle to show a clear insight to your methodology.

Please reformulate the conclusions completely. Use bullets or number and expand them point-by-pint. The introduction used in the conclusions section must be revised.

And NOVELTY??????

Author Response

(The authors gave the same response as above.)

Reviewer 4 Report

Thanks to the authors for their investigation on durability of steel fiber reinforce concrete under environmental effects. The following comments need to be addressed:

·         In line 36 to 38 please refer to the following recent studies:

·         1- Concrete Degradation Due to Moisture and Low- And High-Temperature Cycling. ACI materials journal, 2020, 117 (1), p.129-138.

·         2 - Impact of Laboratory-Accelerated Aging Methods to Study Alkali–Silica Reaction and Reinforcement Corrosion on the Properties of Concrete. Materials, 2020, 13 (15), 3273.

·         In line 88 to 90 add a figure showing the particle size distribution of coarse and fine aggregates.

·         Please specify the type of utilized cement.

·         Provided the fiber characteristics in a table rather than the text in lines 91 to 93.

·         Provide a clear text matrix in a table to clarify the information in lines 100 to 108.

·         What is the benefit of Figure 2? It just shows three cylinders from outside.

·         Very Important: Please add six sub-sections under ‘2.2. Experimental method’ and explain each of the six environmental conditions in detail.

·         What is the freeze cycles temperature range and how many cycles are performed with what duration?

·         Please provide the standard deviation of the compressive strength results.

·         Please add bullet points in Conclusion and provide the main quantitative findings.

·         Please add a paragraph to the Conclusion and explain the limitations of this study and make suggestions for future studies.

Author Response

(The authors gave the same response as above.)

Reviewer 5 Report

The paper describes an interesting study of the influence of the content of steel fibers in concrete on its durability. In doing so, the authors mainly investigated the influence of different environments on the occurrence of microcracks and concrete strength. I think that the topic is relevant and suitable for publication in Materials. However, before accepting, I recommend that the authors improve the contribution in accordance with the following comments:

1. In the state-of-the-art review, I propose to conduct a review of similar methods of characterizing concrete for wear. An example of wear measurement is published in “Three-Dimensional Characterization of Concrete’s Abrasion Resistance Using Laser Profilometry”, https://doi.org/10.5545/sv-jme.2015.2430

2. It is not clear from the description of the experimental methodology whether the specimens are cylindrical or cube-shaped.

3. Use a “digital optical microscope” instead of an “electron microscope”. Also, add its specifications and manufacturer.

4. Explain the “position 1… 3” locations in more detail. By what logic did you choose them and where exactly are located?

5. Table 4 shows the negative correlation of the content of steel fibers in the case of NaCl and Na2SO4 solution. This is an important measurement result, which is not adequately highlighted and discussed.

6. Name of chapter 3.3. is inappropriate and potentially misleading. Under this title, physical modelling such as stress and deformation analysis are often described.

7. The sentence at the end of chapter 3.3 is not clear: “According to this formula, the optimum content is shown in Table 4.”. Which formula? Table 4 does not show optimum content. Please explain.

8. Chapter 3.4 needs to be significantly improved in order to understand what the results in the tables represent. I suggest that you shorten the introductory part and cite the methodology accordingly. Further, explain the significance of the results in Tables 5-8. It is not possible to understand the results in this form.

9. According to the results and discussion, support the last sentence in 3.4 “The results of grey correlation analysis verify the microscopic observation results, and the freeze-thaw cycle has the most significant impact on the failure of concrete specimens.”

Author Response

(The authors gave the same response as above.)

Round 2

Reviewer 1 Report

ok with me

Author Response

(The authors gave the same response as above.)

Reviewer 3 Report

All comments were well considered but no feedback was received regarding the novelty.

Author Response

(The authors gave the same response as above.)

Reviewer 4 Report

The authors have improved their manuscript by trying to address all the comments.

Author Response

(The authors gave the same response as above.)

Reviewer 5 Report

The authors have adequately improved the manuscript according to most of the comments. However, the 8th comment is still not answered:

8. Chapter 3.4 needs to be significantly improved in order to understand what the results in the tables represent. I suggest that you shorten the introductory part and cite the methodology accordingly. Further, explain the significance of the results in Tables 5-8. It is not possible to understand the results in this form.

Author Response

(The authors gave the same response as above.)
